# An Analysis of Multigenerational Issues of Generation X and Y Employees in Small- and Medium-Sized Enterprises in Thailand: The Moderation Effect of Age Groups on Person–Environment Fit and Turnover Intention

**DOI:** 10.3390/bs13060489

**Published:** 2023-06-09

**Authors:** Kanokon Rattanapon, Ann Jorissen, Kevin Paul Jones, Chavis Ketkaew

**Affiliations:** 1International College, Khon Kaen University, 123 Mitraphap Highway, Khon Kaen 40002, Thailand; kanora@kku.ac.th (K.R.); kevinjo@kku.ac.th (K.P.J.); 2Center for Sustainable Innovation and Society, Khon Kaen University, 123 Mitraphap Highway, Khon Kaen 40002, Thailand; 3Faculty of Business and Economics, University of Antwerp, Prinsstraat 13, 2000 Antwerpen, Belgium; ann.jorissen@uantwerpen.be

**Keywords:** Generation X, Generation Y, supportive work environment, turnover intention, person–environment fit, person–supervisor fit, person–group fit, person–organization fit, person–job fit

## Abstract

Multigenerational employees can evidently impact human resource management practices in terms of effective employee retention. Arguably, a high turnover intention of young employees can hinder a company’s human resource development, while a high volume of retirement of senior employees can create a skill deficit and even a labor management dilemma. This study explored how a supportive work environment can retain employees of different age groups in Thailand’s small- and medium-sized enterprises (SMEs), particularly Generation X and Y. It modeled a supportive work environment that impacts the behaviors of Generation X and Y employees, taking into consideration the relationship among factors such as person–job fit, person–group fit, person–supervisor fit, person–organization fit, person–environment fit, and turnover intention. This paper statistically analyzed a set of data drawn from an attentive survey of a total of 400 employees of SMEs in 4 populous provinces in Thailand using structural equation modeling (SEM) and multigroup analysis (MGA) with the moderation effect of generations. This paper then found that person–job fit, person–group fit, person–supervisor fit, person–organization fit, person–environment fit, and turnover intention can influence an employee’s intention to remain in his/her job. Additionally, the relationship manipulation among the aforementioned variables might influence Generation X and Y employees differently. Under the circumstances, supervisory support with less group involvement may encourage the retention of Generation Y employees, whereas a sufficient focus on job suitability could improve the retention of Generation X employees.

## 1. Introduction

Strauss and Howe (2009), in The Fourth Turning: What the Cycles of History Tell Us About America’s Next Rendezvous with Destiny, have proposed generational theory that divides people into “age cohorts” or generational groups based on the changing phrases of global economic and technology evolutions [1]. According to Strauss and Howe’s generational theory, people in the same cohort share interests, values, and even characteristics.

Generally, the term Generation X people (or Gen-X) refers to those born between 1965 and 1980 in an era of wealth. They have lived comfortably and grew up with the development of computers, video games, and hip-hop music. This was when the birth rate was controlled relative to the Baby-Boomer generation [2]. As Generation X is closer to the Baby Boomers, the characteristics of the Baby Boomers also apply to a large extent to Generation X. The older generation (Generation X) is less individualistic, accepts more hierarchical forms of leadership, respects more authority, prefers life-long employment, and appreciate more consensus than younger generations [3]. On the other hand, Generation Y (or Gen-Y) includes those born between 1981 and 1995. Gen-Y people grew up with digital technology, have had a convenient life, and were born in the era of economic growth. So, Gen-Y people tend to have good educational opportunities, possess strong ideas about what they like, and reject things they do not like [4]. Generation Y emphasizes independence, autonomy, job content, and work–life balance [2,5]. Generation Y also prefers less hierarchical forms of leadership [3]. Gen-Y employees struggle to raise their children and keep their professions. They are worried about their elderly parents and uncertain future career due to a global economic downturn [6]. Some Gen-Y workers are likelier than workers from older generations to report being less productive and less connected to their employers [7]. According to research undertaken in Thailand, Gen-Y employees are likelier to abandon their jobs [8,9,10].

Human resources as an essential component of organizational management can yield more challenges to small- and medium-sized enterprises (SMEs) than others. It is crucial that managers of SMEs are able to effectively address potential challenges in their human resources (e.g., differences between the people within the organization). In this regard, the issue of age diversity is among various challenges that can impact human resources management, as people of different generations possess different characteristics, attitudes, values, needs, and approaches to either their private or professional lives. Additionally, employees of different generations might find different motivations to keep their job positions and stay competitive at work. For instance, the study of Slovak employees, M. Hitka et al. (2021) finds that the rate of base pay serves as the most effective motivation for employees, such financial motivation has a varied degree of impact on employees who hold different gender, age, and job positions [11]. In this case, blue-collar males and white-collar females are more motivated by income rate than other groups.

However, in the case of Thailand, despite facing high financial challenges due to the country’s economic issues and political instability, in the past recent years, Thailand witnessed a high resignation rate in its labor market. Arguably, the number of employees quitting their employment has increased significantly since 2021, particularly in the SME section. (Napathorn, 2022) Evidently, most businesses in Thailand hire multigenerational employees [8]. Statistically, those businesses consist of a large proportion of Generation X (around 50 to 60%) and Generation Y (around 40 to 50%), and a small proportion of Generation Z (less than 10%) [12]. It is of this paper’s interest to explore factors influencing a supportive work environment and turnover intention. In this research, an emphasis is placed on the turnover intention of Generation X and Y employees, as they are the majority of workers in Thailand. Consequently, retiring Gen-X employees and the high turnover of Gen-Y employees in Thailand raises the question of how might Thailand’s SMEs deal with the talent shortage problem. It is crucial that human resource managers understand how to manage different generations properly to ensure a good retention rate of efficient employees. 

Recent studies indicated that supportive work environments could help to reduce turnover intention [13,14,15]. This research hypothesized that creating supportive work environments as a human resource management approach helps to alleviate turnover intention while effectively dealing with generational diversity in an organization. Accordingly, this study also hypothesized that the employee generation (Gen-X and Gen-Y) moderates the relationship between person–environment fit and turnover intention. 

As for theoretical perspectives, it is critical to note that studies on generational diversity, supportive work environments, and turnover intentions are scarce and controversial in many cases. It is important that researchers and practitioners should therefore be cautious about generalizing the results of previous studies [14,16,17,18] to their research or workplace settings. As a result, an empirical study design is needed if one is to investigate generational diversity and human resource management in other circumstances. Additionally, recent studies rarely explore the moderating role of generation (Gen-X and Gen-Y) on the relationship between a supportive work environment and turnover intention. Instead, some topical research papers focused on the moderating role of other variables on the relationship between person–environment fit and turnover intention. Those variables were non-demographic variables that included person–job fit [19], grit [20], psychological empowerment [21], organizational innovation climate perception [22], organizational ethical standard [17], temporariness dimensions of the job environment [23], and psychological safety [24]. Howard and Cogswell (2022) explored the dimension of person–environment fit using employment duration (years) as a moderating variable [15]. Yet, the employee generation was not applied as a moderator in their studies. On the other hand, a recent study regarding turnover intention and employee retention used generations as moderating variables, but supportive work environments were overlooked [25,26]. In sum, none of those papers investigated or described how generational diversity might play a moderating role in the relationships between supportive work environments and turnover intention by targeting SME employees. The aim of the present study was to bridge this knowledge gap.

This research aims to understand how Generation X and Y employees work differently and to determine how a supportive work environment can affect employees’ turnover intention. This paper’s originality is demonstrated through an application of the conventional SEM relationship between supportive work environments and employee turnover intention by using generation (Gen-X and Gen-Y) as a moderating variable with the data collected. The present study adds to the existing body of knowledge by neatly addressing this knowledge gap. We did so by having the generation variables (Gen-X and Gen-Y) play a moderating role in the SEM model, and our results demonstrated a relationship between a supportive work environment and employee turnover intention. The framework primarily employed in this study was adopted from person–environment fit theory [27], which tries to explain the relationship between individual qualities and the work environment. This theory is suitable because it presumes that a good match between individuals and their environment will lead to a lower intention to leave [28]. Additionally, the congruence may be researched at the individual, group, and organizational levels, offering a wide range of research applications. Based on the person–environment fit theory, we also hypothesize that person–supervisor fit, person–group fit, person–organization fit, and person–job fit would raise person–environmental fit and would finally reduce turnover intention [16,28,29]. The results of this study will improve the ability of employers to retain Generation X and Y employees via a supportive work environment.

## 2. Review of the Related Literature and Theoretical Development

Efficient employee management directly contributes to the performance of an organization [30]. The influences impacting a worker’s decision to stay or quit a firm are critical to the organization’s competitiveness. If a company wants to keep its employees, it must first understand the reasons that shape employees’ decision to leave. 

Generation X and Y employees are considered the primary workforce in the developing world. However, very few research papers demonstrated the relationship between generational differences (between Gen-X and Gen-Y employees) and turnover intention. Roman-Calderon et al. (2019) studied how behavioral variables, such as professional respect and job empowerment, influence employee turnover intention using Generation X and Y workers as a sample from a developing country, Columbia [31]. They found significant differences between Generation X and Y behaviors, but it did not highlight the role of a supportive work environment. However, the turnover intention was not directly linked to this structural model. Another research article examined generational changes in work attitudes across three generations, focusing on the similarities and differences between Gen-Y employees and previous generations [32]. Gen-Y reported lower levels of overall job satisfaction, work engagement, and organizational trust compared to Boomers and Gen-X employees. Although many variables were linked to turnover intention, person–environment fit was not involved in the research model. Moreover, generational differences were included as potential moderators in recent studies that investigated turnover intention and employee retention; however, supportive work environments were not the primary focus of this investigation [25,26]. Additionally, a study in Serbia aimed to explore the link between flexible working arrangements and employee turnover intentions with job satisfaction serving as a mediator [33]. The findings indicate that flexible work arrangements may lead to increased job satisfaction, which in turn leads to decreased turnover intentions. However, the effect of generational diversity was not taken into account. Nevertheless, a study conducted in Gauteng, South Africa, suggested that issues related to generational diversity in the workplace are inconclusive [18]. This paper studied the relationships between work engagement and generational diversity and revealed no substantial discrepancies between Generation X and Y’s work involvement. Taken together, information regarding how generational diversity may influence job turnover remains scarce. An empirical study is therefore needed to investigate the impact of generational diversity on human resource management in a specific geographical region, such as Thailand.

The following sections review and discuss related research articles explaining how work environment may influence an individual’s turnover intention and the moderating role of generational diversity. These relationships help to establish the research hypotheses and formulate the research framework.

### 2.1. Person–Environment Fit (PEF) Theory and Turnover Intention (TI)

The association between the person and the environment can affect human behavior [34]. Person–environment fit (PEF) can be defined as the comfort or the perceived compatibility of employees with the environment [14,27,35,36]. PEF can happen when there is compatibility between the employee’s needs and the environment [37]. PEF theory explains the relationship between an individual’s traits and his or her environment in which the individual impacts his or her surroundings while likewise being influenced by it [38]. The person’s motivation, behavior, and overall mental and physical health are all affected by the fit between the person and the environment. As a result, if the fit is appropriate, the individual’s performance may improve. The individual may develop maladaptation if the fit is incompatible.

Employee turnover refers to the rate at which employees leave an organization [39]. Employees can leave an organization voluntarily or involuntarily. The theory of planned behavior (TPB) demonstrates that an individual’s intention may regulate his or her behavior [40]. Hence, intentions can be considered a suitable predictor of behavior [16,17,20]. Many studies have focused on intentions instead of quitting behavior due to the complexities of following up with employees during their careers [41,42], and some have reported a positive connection between turnover intention and actual turnover [43,44]. Therefore, employee turnover intention is an essential factor that organizations use to predict employee turnover [45]. Takase et al. (2005) supported that the better the PEF, the greater the employee’s intention to remain in a job [42]. However, the worse the PEF, the more likely the employee will resign.

According to Liao, P. Y. (2022), Abbas et al. (2015), Krishnan et al. (2017), and Ketkaew et al. (2020), PEF has several dominant themes, including person–supervisor fit, person–group fit, person–organization fit, and person–job fit [16,28,35,41]. The variables and their relationships are stated in the following sections.

### 2.2. Person–Supervisor Fit (PSF)

Person–supervisor fit (PSF) describes the relationship between two employees (where one is the supervisor of the other). Although this relationship can develop in many ways, such as between co-workers [46], recruiters and applicants [47], and mentors and protégés [48], the majority of this field’s research focuses on the connection between leaders and subordinates [28,35,49,50,51]. The related studies in this area cover the consistency of the relationship between leaders and followers [52,53], similar personalities between supervisors and subordinates [54], and the consistency of the goal between managers and employees [55]. In each circumstance, the characteristics of supervisors can indicate the environment. Based on the literature, supervisors may also embrace and demonstrate the attributes of person–group fit (PGF) and person–organization fit (POF) to subordinates [56,57]. The compatibility between employees and their supervisors, which influences PEF, is referred to as person–supervisor fit (PSF). Subordinates tend to be happy and privileged when developing good relationships and having positive interactions with their supervisors, resulting in improved PEF [28]. However, Ketkaew et al. (2020) suggested PSF does not associate with PEF regardless of age levels [16]. With the inconclusive results of the literature, we decided to maintain the original relationship proposed by Abbas et al. (2015) and PEF theory [38]. Thus, this research hypothesizes that PSF is positively correlated to PEF (H1).

**Hypothesis** **1** **(H1).**
*Person–supervisor fit (PSF) will positively affect person–environment fit (PEF).*


### 2.3. Person–Group Fit (PGF)

Person–group fit (PGF) refers to the compatibility between persons and the workgroup [35,38,58,59]. Although many people are interested in co-workers’ similarities based on their demographic variables [60], it is important to focus on how compatibility among co-workers can influence outcomes in group settings. Compatibility includes personality, the style of work [61], value [62], and goals [63]. The relationship between people and the workgroup facilitates the employees to work together smoothly. PGF divides personnel into workgroups, allowing certain team members’ skills to balance the shortcomings of others [28]. Individuals can communicate more readily with other members of a team if they have a positive relationship with them, which leads to greater performance. According to a study on PGF, hiring people with similar personalities improves employee communication and stimulates social engagement, which has a positive impact on PEF [28]. In addition, Ketkaew et al. (2020) found no relationship between PGF and PEF [16]. With indecisive results about this relationship, H2 hypothesizes that PGF is positively associated with PEF, maintaining Abbas et al. (2015)’s original argument and PEF theory [38].

**Hypothesis** **2** **(H2).**
*Person–group fit (PGF) will positively affect person–environment fit (PEF).*


### 2.4. Person–Organization Fit (POF)

POF compares the personality, goals, and values of the employees with those of the organization [27]. POF mainly focuses on needs and values [38]. POF demonstrates that improving the relationship between employees and the organization will raise the likelihood that they will want to stay. POF is based on the congruence of the value and goal [27]. The congruence of the value means the employees and organizations have similar values [27,35,64]. Empowering an individual’s values may also enhance organizational values and culture [65]. Suppose the employees have a value that is congruent with the organization. In that case, they will create a positive attitude towards the organization [66], and there may be a higher chance for the employees to stay with the organization [67,68]. However, the congruence of the goal means a similar boundary between employees’ goals and the organizations’ goals [69]. These similar goals can attract employees to the organization. Following POF and based on PEF theory [38], a firm will hire people who share the same values as their current employees, resulting in a more favorable PEF [16]. In this research, H3 hypothesizes that POF is positively associated with PEF.

**Hypothesis** **3** **(H3).**
*Person–organization fit will positively affect person–environment fit (PEF).*


### 2.5. Person–Job Fit (PJF)

Person–job fit (PJF) is the compatibility between employees’ characteristics and their jobs [27,70]. PJF has two basic concepts [70]: the demands–abilities fit, which measures how well the skills, knowledge, and ability of employees match the job requirement, and when the job and its characteristics can meet the employees’ desires, needs, and the preferences of the employees. PJF focuses on satisfaction, adjustment, and well-being [35,71,72]. This integration aids in determining employee suitability with the duties they perform within the firm, leading to the congruence between a person and his/her associated work environment [16]. Based on PEF theory [38], hence, H4 hypothesizes that PJF is positively associated with PEF.

**Hypothesis** **4** **(H4).**
*Person–job fit will positively affect person–environment fit (PEF).*


In sum, PSF, POF, PGF, and PJF positively affect PEF. Based on Takase et al. (2005) and Ajzen (2002) discussed in the PEF section, the more PEF, the better intention an employee will remain in a job, on the flip side, reducing an employee turnover rate [40,42]. Therefore, H5 hypothesizes that PEF is negatively associated with turnover intention (TI).

**Hypothesis** **5** **(H5).**
*Person–environment fit will negatively affect turnover intention (TI).*


### 2.6. Moderating Roles of Generational Differences

Age may affect an employee’s behavior [6,17,32,73]. Generational diversity at work may affect an individual’s job performance in several ways. Among the empirical studies of a correlation between age and employee behaviors, there are both advantages and shortcomings of dealing with multigenerational employees in human resources management. According to Vraňaková et al. (2021), older generation employees tend to promote their competitiveness at work and support the younger generation in sharing their knowledge and experiences [74]. The younger employee behaviors then are encouraged by the quality work environment. In a similar light, Celik et al. (2021) propose that Gen X and Gen Y employees show their commitment and performance to their job supervisors to different degrees. In this regard, Gen X employees have a stronger commitment to job performance than Gen Y employees who are more proficient in integrating information technology into their job performance. As a result, Celik et al. (2021) suggest that Gen X and Gen Y employees need supervisors with different leadership styles [75]. A recent study using binary logistic regression and neural network analysis to analyze the impact of age characteristics and behaviors of Gen X and Gen Y employees shows that Gen Y employees in the private business sector in Sri Lanka have a higher turnover rate than Gen Y employees [76].

In this respect, this paper then hypothesizes that there may be differences in behavior between Generation X and Y regarding how PEF may influence turnover intention. In the association between PEF and turnover intention, age plays a moderation effect. Using generation as a moderator in the SEM approach helps to reduce possible measurement errors and biased results that may occur in the standard regression and analysis of variance approaches [77]. The proposed structural model shown in Figure 1 is presented according to the stated hypotheses.

## 3. Methodology

This section explains the approach we used to conduct this research, including (1) Measures, (2) Participants, Sampling, and Data Collection, and (3) Data Analysis. Structural equation modeling (SEM) and multigroup analysis (MGA) techniques were used to explore potential associations between supportive work environment characteristics and turnover intention variables. Since we are attempting to develop a multifactor model to predict the turnover intention of a cross-sectional sample separated into various groups, these methods are appropriate for this study [78].

### 3.1. Participants, Sampling, and Data Collection

This research employs the quota sampling approach. Based on the recent data obtained from Open Government Data of Thailand (2021), the finite population of employees working at registered SMEs in Thailand is 1,229,347. The companies’ sizes range from 1–249 employees [79]. Among 109,582 SMEs in Thailand, the highest number is in wholesale business (19,419 firms), the second is in retail business (14,855 firms), the third is in food production business (10,163 firms), and the fourth is in agriculture (5027 firms). Among all companies, the majority are considered micro (60,619 firms), the next is S (35,917 firms), and followed by M (9097 firms). 

We used Calculator.net to calculate the appropriate sample assuming normal distribution with a margin of error of 5% and a confidence level of 95%. This result suggested 273 or more surveys are needed. Additionally, Kline (2016) recommended a sample size of 385 based on the SEM analysis [80]. When a structural model contains 7 or fewer constructs, Hair et al. (2010) advised a minimum sample size of 300 [77]. Since there is no commonly accepted rule in determining sample size in a statistical analysis using SEM, given 6 constructs (latent variables) in total, the researchers decided to use a sufficient sample size of 400 respondents following the minimum allowable amounts by Kline (2016) and Hair et al. (2010). 

The plan was to collect data from 400 SME employees working in Bangkok, Chiangmai, Khon Kaen, and Phuket provinces, the capital and most populous cities representing 4 regions of Thailand. The locations included urban office areas and remote agricultural areas. The respondents were SME employees from wholesale, retail, and food service businesses. The sample was divided into mutually exhaustive subgroups using an uncontrolled quota sampling strategy based on geographical locations. In each province, the data from 100 employees were collected using a questionnaire, making up a total of 400 employees (Bangkok 100, Chiangmai 100, Khon Kaen 100, and Phuket 100). An equal number of surveys was also implemented in a study in the human resources management area by Islam et al. (2020). According to the National Statistical Office (2022), the proportion of Gen X to Gen Y (X:Y) employees in the selected provinces is around 60:40 [12]. Therefore, the quotas for X and Y are 260 and 140 respondents, respectively.

As for data collection, we used the intercept survey method which allows an informant to complete the survey in one setting, resulting in better response quality due to less distraction and interference [78,81]. When employing the intercept survey approach to obtain on-site perception information from respondents in public settings, this method is appropriate because equal weights (n = 100) represent equal importance designated to each location [78,81]. The research team approached anonymous office workers, both Gen X and Y, passing by the main halls and reception areas of different office buildings [82]. The data were collected during the post COVID-19 era. Before responding to the questionnaire, participants were made aware of the importance of anonymity as well as the ethical concerns of business and social science research. To ascertain whether the workers work in that specific area, a screening question was asked. If the answer was “Yes”, the enumerators would keep allowing those office workers to complete the surveys. The enumerators would stop the procedure if the answer was “No”.

Due to many factors, there may be method bias in common scale attributes during the data collection process [83]. This issue was solved in this study by notifying respondents that, while some questions may appear to be similar, they are all different in important ways. The respondents were invited to thoroughly read each item. The surveyors were able to collect the data with the demographic characteristics shown in Table 1 of the Results section. With time and budget constraints during data collection, the collected data based on age criterion only reveals Gen X of 294 employees (73.50%) and Gen Y of 106 employees (26.50%). According to the quota proportion, the achieved Gen X respondents were more than the planned number (294 > 260), but Gen Y respondents were less (106 < 140). However, we decided to continue the analysis because the data met the geographical location quota (100 respondents from the 4 provinces) and the normal distribution criterion. 

### 3.2. Measures

Based on the proposed model (see Figure 1), we developed a questionnaire that could be answered in either English or Thai. There are two parts to this questionnaire. Part 1 is made up of general questions, including three multiple choice questions asking about respondents’ demographic profiles, including gender (male and female), generation (Generation X, 43 to 57 years old, and Generation Y, 27 to 42 years old), and education (high school, bachelor’s degree, graduate degrees, and diploma). In Part 2, six constructs represent the latent variables and their measures (see Appendix A). The perception questions are presented on a five-point Likert scale, which gives respondents ranking alternatives for how much they agree or disagree with a statement. As for the scales, 1 represents the lowest level, and 5 represents the highest, with the neutrality or uncertainty level represented by the midpoint (3).

For this questionnaire, the indicators used to measure each construct were explored and identified during the literature review. According to Appendix A, the scales and measures in this questionnaire were adopted from Ketkaew et al. (2020) whose research reexamined the model that was originally validated by Abbas et al. (2015) [16,28]. 

### 3.3. Statistical Procedure and Analysis

Prior to performing the SEM analysis, we conducted the following statistical tests to validate the data and identify possible biases resulting from the data collection. First, the common method variance (CMV) of the gathered data was investigated using Harman’s single-factor test [84]. CMV creates a systematic covariation above the genuine relationship between the indicators (observed variables), which may result in false estimates of the magnitude and significance of the relationships among constructs (latent variables). When the indicators were subjected to Harman’s single factor test, the findings demonstrated a cumulative variance of 23.167 percent (less than the 50% threshold), confirming the absence of CMV. Deficient CMV implies a zero likelihood for cross-loadings between indicators for different constructs. 

Next, the Cronbach Alpha test was performed to test the reliability of the questions in this questionnaire. The results are all acceptable because all the values pass the threshold of 0.70, indicating the internal consistency of the subquestions in each construct. Additionally, the basic assumptions underlying the SEM approach must be satisfied prior to the main analysis. Before proceeding with hypothesis testing, the data were checked for multicollinearity, normality, outliers, and missing values as Ahman et al. (2021) suggested [85]. 

In this research, the major data analysis method used was the SEM approach. We applied the two-step approach for SEM as suggested by Anderson and Gerbing (1998) [86]. These two steps include (1) examining measurement model validity using confirmatory factor analysis (CFA) and (2) checking the structural model goodness of fit. Step 1 is to test the validity and reliability of the outer (CFA) model by measuring the association between each indicator and its variables. The goodness of fit (GOF), convergent validity, and discriminant validity are all examined in this step. The GOF was tested by comparing the calculated values with the fit indices thresholds: chi-square/df ≤ 5.00, CFI (comparative fit index) ≥ 0.90, IFI (incremental fit index) ≥ 0.90, TLI (Tucker Lewis index) ≥ 0.90, and RMSEA (root mean square error of approximation) ≤ 0.08. Convergent validity was assessed using the construct’s average variance extracted (AVE) values and the composite reliability (CR). AVE and CR thresholds are 0.50 and 0.70, respectively [87,88]. The Fornell and Larker (1981) criteria were used to assess discriminant validity [87]. Step 2 evaluates the inner structural model to see if the entire model is reliable, applying the fit indices thresholds similar to the CFA approach. 

Next, this study examined the moderation effect of Generation X and Y on this structural model using multigroup analysis (MGA). The sample was divided into Generation X (43–54 years old) and Generation Y (24–42 years old). In this step, the measurement invariance (MI) test and the z-test were performed to identify differences in behaviors between these two groups. Using the multigroup analysis (MGA) technique, MI is a method for determining whether a measurement significantly differs between two groups (generations X and Y) [89]. The MI technique establishes configural invariance (unconstrained model), metric invariance (equal factor loadings), and scalar invariance (equal intercepts), which are all based on the CFA model. According to Steenkamp and Baumgartner (1998), partial MI is supported when only configural and metric invariance is met [89]. Full MI, on the other hand, holds when partial MI and scalar invariance are met. Then, using the GOF criteria, we looked at the consistency of the structural models in which generation X and Y subgroups were simultaneously estimated. The critical ratio for path differences, adopted from Byrne (2010), was then determined using the MGA technique to compare the factor loadings of the models between two groups [90]. The z-test approach for critical ratio differences was applied to examine statistical differences between Generation X and Y’s factor loadings [90]. Gao et al. (2008) suggested a threshold of 1.96 when a sample with a multivariate normal distribution was obtained [91], which is consistent with the SEM assumption [77]. 

## 4. Results

This section involves the respondents’ profile and SEM results, including the measurement model, structural model, and multigroup moderation analysis. Table 1 presents the demographic profile of the study’s respondents. Of the total 400 respondents, 58.50% are female. As for the majority, Generation X people accounted for 73.50%, and the participants with bachelor’s degrees accounted for 58.75%. 

All the basic SEM assumptions (multicollinearity, normality, outlier, and missing values) were satisfied [85]. From our careful data collection process, 400 observations revealed neither missing values nor outliers (employing the stem-and-leaf method) [92]. In addition, all correlation coefficients were less than 0.85 [93], implying the absence of multicollinearity among the considering variables. Lastly, the normality criterion was fulfilled since the skewness and kurtosis values were acceptable, as suggested by Byrne (2010)—between ±1 for skewness and ±3 for kurtosis [94]. 

There are two main steps to analyze the SEM results: Step 1. measurement model, and Step 2. structural model [86]. Moreover, the multigroup moderation analysis of generational diversity is provided in Step 3. The preliminary results section involves Step 1. In addition, the primary results and discussion section involves Steps 2 and 3.

### 4.1. Step 1: Measurement Model (Confirmatory Analysis)

To test our measurement model, we employed confirmatory factor analysis (CFA). Hair et al. (2008) suggest that the CFA approach involves evaluating many criteria, such as internal consistency, reliability, convergent validity, and discriminant validity [95]. All the constructs were connected using the covariances before we started testing. To improve the goodness of fit, error terms in the same construct were allowed to correlate [95]. 

#### 4.1.1. Goodness of Fit

In Table 2, we show the goodness of fit of the measurement model. The result was acceptable for a complex model. The measure of all the fit indices passed the stated thresholds: the chi-square/df is 2.163, the CFI (comparative fit index) is 0.945, the IFI (incremental fit index) is 0.945, the TLI (Tucker Lewis index) is 0.936, and the RMSEA (root mean square error of approximation) is 0.055. Hence, this satisfied the goodness of fit criterion.

#### 4.1.2. Convergent Validity

Based on average variance extracted (AVE) and composite reliability (CR) properties, convergent validity investigates internal consistency within each construct. Convergent validity was determined by comparing AVE and CR to the predefined thresholds of more than 0.50 [87] and more than 0.70 [88], respectively (see Table 3). 

According to Table 3, the results show the PSF, PGF, POF, PJF, PEF, and TI variables meet the criteria. For the PSF variables, the *p*-values of all indicators are lower than 0.001, and when we compared this with the threshold, both AVE (0.637) and CR (0.875) were higher. Moreover, for the PGF variables, all indicators’ *p*-values are lower than 0.001, and both AVE (0.507) and CR (0.804) are higher than the threshold. Similar to the POF variables, the *p*-values of all indicators are lower than 0.001, and both the AVE (0.621) and CR (0.866) are higher than the threshold value. Like the PJF variables, the *p*-values of all indicators are lower than 0.001, and both AVE (0.673) and CR (0.892) are higher than the threshold. However, for the PEF variables, AVE (0.497) is lower than the threshold, while CR (0.796) is higher, thus considered acceptable. For the TI variables, the *p*-values of all indicators are lower than 0.001, and both AVE (0.721) and CR (0.911) are higher than the threshold. All in all, the *p*-values of all indicators are lower than 0.001, which shows that all constructs have indicators that converge significantly to the measurement model derived from the literature review.

#### 4.1.3. Discriminant Validity

Discriminant validity is the degree of the differences between two or more conceptually similar constructs, and it also tests the relationship between each construct. Possible cross-loadings amongst the variable measures for various latent constructs can be identified in this process if the discriminant validity condition is not satisfied. In this part, we compared the square root of AVEs (see diagonal bold texts in Table 4) with the correlations in the associated matrixes (both vertically and horizontally) [87]. Table 4 shows that the degree of the differences between each construct had passed this validity check, and we can disregard the issue related to cross-loadings between different constructs.

The next sections explain the results of the structural model and the multigroup moderation analysis involving measurement invariance analysis and loading differences tests. Discussion regarding the loading differences between Generation X and Y is also presented. 

### 4.2. Step 2: Structural Model

A structural model was created after checking the measurement model by linking all the constructs following the proposed model demonstrated in Figure 1. To validate the structural model, several of the goodness of fit indices should exceed the specified thresholds [96]. After running all variables through the structural model, the goodness of fit results (see Table 5) show that the chi-square/df (2.246) is lower than 3.00, the CFI of 0.940, IFI of 0.941, TLI of 0.931 were higher than 0.90, and the RMSEA of 0.057 was less than 0.10. This could be summarized by the fact that the goodness of fit indices were in the satisfied range.

According to Table 6, the results show that the hypotheses we created had some conflicting results. Although the hypotheses’ test results supported H3, H4, and H5 at a significance level of 0.001 and supported H1 at a significance level of 0.05, H2 (that PGF is positively associated with PEF) was rejected. As for H1, this result supported Abbas et al. (2015) [28] but was inconsistent with Ketkaew et al. (2020) [16], demonstrating that more PSF significantly improves PEF. Subordinates could feel more secure when their supervisors support them both physically and emotionally, resulting in a pleasant work environment. In line with Ketkaew et al. (2020), the test results also support H3, which means POF positively influences PEF [16]. Clearly, the work environment could be enhanced when an employee’s goal is consistent with the organizational value. An argument by Ketkaew et al. (2020) also reinforces H4 [16]. PJF significantly improves PEF when an employee’s qualifications match well with specific job requirements. 

However, the test results were contradictory to Abbas et al.’s (2015) findings on which H2 was based [28]. Rejecting H2 implied consistent results with Ketkaew et al. (2020) in that PGF did not positively correlate to PEF in the circumstance of Thailand [16]. Conversely, the research findings revealed that increasing PGF may reduce PEF and vice versa. The fact that Bangkok, Chiangmai, Khon Kaen, and Phuket are cities with urban citizens striving to survive in a competitive environment may increase a sense of individuality in society. An employee feels more relaxed and productive when working independently. Working as an individual and spending less time with co-workers during work hours may enhance PEF, thereby reducing an individual’s turnover intention. 

### 4.3. Step 3: Multigroup Moderation Analysis

#### 4.3.1. Measurement Invariance Analysis

Measurement invariance (MI) is a method that can be used to check whether a measurement method is understood differently between two groups [89]. Here, MI was used to ensure that the respondents (Generation X vs. Y) understood the questions in the questionnaire in a similar manner. According to the CFA approach, three models need to be evaluated to satisfy measurement invariance analysis. The first is the unconstrained model, called configural invariance. The second is to load the equal factors, which is called metric invariance. The third is the equal intercepts, which is called scalar invariance. If only configural and metric invariance tests are satisfied, the model can support partial measurement invariance. Then, we can compare the factor loadings between the two groups. However, if we want to establish the full measurement invariance, we need to hold the partial measurement invariance and also achieve scalar invariance. Finally, the full measurement invariance allows us to evaluate the factor loadings between the groups. Table 7 shows the MI tests after the CFA model.

The thresholds for configural invariance, metric invariance, and scalar invariance were all reached, and the results were accepted (Table 7). The chi-square/df values of all the invariance models are less than 3; CFI, IFI, and TLI of all the models are more than 0.9; and the RMSEA of all the models are less than 0.1. Therefore, in the next part, full measurement invariance can be used to compare the factor loadings between the groups (Generation X and Generation Y).

#### 4.3.2. Z-Test for Loading Differences between Generation X and Y

For this part, we proceeded with a z-test to compare the factor loadings between Generation X and Generation Y using the critical ratio differences [97]. This approach provided the pairs of standardized loadings of Generation X and Y for the hypothesized relationships and the test results of critical ratio differences. If the critical ratio difference is more than the threshold (the absolute value of 1.96), we can conclude a significant difference in factor loadings between the two groups. Table 8 indicates that only the critical ratio difference of path H1 of 2.052 is statistically significant, while those of paths H2, H3, H4, and H5 are insignificant.

## 5. Main Discussion

This section discusses the SEM and multigroup analysis results. According to Table 8 and Figure 2, the standardized loading between Generation X and Generation Y of H1 has a critical ratio difference of 2.052 at a significance level of 0.05. The results of H1 show that PSF does not affect PEF for Generation X (loading = 0.067 and insignificant). But for Generation Y, PSF does affect PEF (loading = 0.329 and *p*-value < 0.05). It was clear that supervisor support was still required for the young generation to adapt well to the work environment and related work tasks, supporting Abbas et al. (2015) [28], Cox et al. (2014) [3], and Smola and Sutton (2002) [5]. Generation X workers (43 years and above) required less attention from their supervisors because many of them were already senior employees of their organizations. When Generation X workers were separately analyzed, the insignificant relationship between PSF and PEF strongly supported Ketkaew et al. (2020)’s argument [16]. In short, it found that Generation X has less of a need for a supervisor compared to Generation Y. This might be because Generation X employees are older. Their position in the organization is likely to be higher, and thus, they might have a direct supervisor. However, Generation Y employees are younger, so they might not yet fully understand their work and, thus, still require guidance after the health crisis.

For H2, the standardized loading difference between Generation X and Generation Y is insignificant (critical ratio difference of |−1.518| < |−1.96|). Consistent with Ketkaew et al. (2020) [16], this finding shows that PGF does not affect the PEF for generation X (loading = −0.139 and insignificant). But for Generation Y, there is a negative effect between PGF and PEF (loading = −0.304 and *p*-value < 0.05). This new relationship demonstrates a novel finding. As previously discussed, young workers are more comfortable working as individuals and feel less attached to the group while at work. This value may positively affect an individual’s person–environment fit and reduce turnover intention. The results infer that Generation Y employees have less of a need to work in a group than Generation X, and Generation Y employees prefer to work individually. Remote work tends to be more appropriate for Generation Y and Generation X. Generation Y people prefer to work by themselves because working in a group does not help them work better, there will be more problems when working in groups, and they cannot make a final decision quickly. 

The result of H3 shows that POF affects PEF for both generations (For Generation X, loading = 0.401 and *p*-value < 0.001; for Generation Y, loading = 0.651 and *p*-value < 0.001). However, the loading is not statistically different (critical ratio difference of 1.156 < 1.96). This result implies that increasing person–organization fit might positively influence person–environment fit and reduce an individual’s turnover intention for both generations [16,28]. Thus, the organization and its employees have to have similar personalities, values, and goals, which can help the employees to work happily within the organization.

As for H4, PJF positively affects PEF only for Generation X with a loading of 0.461 (significant at <0.001) but not for Generation Y. When a Generation X employee feels that job assignments are ample, his or her perception of the work environment becomes positive [16]. However, the association between PJF and PEF cannot be demonstrated by Generation Y’s behavior, considering the research novelty. This research shows Generation Y workers are less attached to PJF but place more importance on POF. For Generation Y, the decision to leave or stay depends highly on the compatibility of individual and organizational values. Additionally, the critical ratio difference between Generation X and Generation Y is −1.474 and insignificant, indicating that their loadings are indifferent statistically. This finding implies that for Generation X, the strongest determinant of fitness to the work environment is person–job fit, which in turn dictates an individual’s decision to quit his/her job. Mediated by person–environment fit, the more degree of person–job fit, the less likely an individual in Generation X will leave the job. Hence, the findings reveal that Generation X employees need to work in a job that matches their characteristics, whereas Generation Y employees do not. This finding demonstrates that a Generation X employee emphasizes a job’s suitability and wishes to remain in that job for the long term. However, a Generation Y employee does not give importance to whether his/her personality, skills, or knowledge are fitted with the job and may leave that job regardless of the job suitability. 

For H5, the result of H5 showed that PEF affects turnover intention negatively for both generations (for Generation X, loading = −0.419 and *p*-value < 0.001; for Generation Y, loading = −0.297 and *p*-value < 0.05). This finding is also consistent with Ketkaew et al. (2020) [16]. However, there was no statistical difference between the standardized loadings of Generation X and Generation Y (critical ratio difference 1.38 < 1.96). Thus, the organizational environment is the main factor that helps employees remain in their job, affecting individual outcomes. If the employees and the environment match, there will be a greater intention to remain in the job. However, if the employees and the environment mismatch, the turnover intention will be high [42].

Finally, the most significant factor that positively influences PEF for both Generation X and Y employees is POF [16]. However, the loadings of both Generation X and Y are not significantly different as the critical ratio difference of 1.156 is less than the threshold of 1.96. The findings show that when personal values are aligned with business values and culture, the work environment can be improved regardless of age. This positive PEF plays a significant role in reducing the turnover intention of both generations [16,42]. However, for Generation X, PSF and PGF do not influence PEF. For Generation Y, PJF does not impact PEF, but PGF has a negative impact on PEF. 

## 6. Conclusions

### 6.1. General Findings

Employee turnover means costs that an organization has to bear. A workplace with multigenerational employees may also face human resources management issues. Retiring Gen-X and the high turnover of Gen-Y employees in Thailand are expected to cause a talent shortage, which is a critical human resources management issue after the pandemic. Understanding the quitting motives of both Generation X and Y allows an organization to manage its work environment to alleviate staff turnover. This research answers the question of how a supportive work environment can help reduce turnover intention among Generation X and Y employees. This study contributes to the human resources management literature by introducing the moderating role of generational diversity to the conventional SEM model of person–environment fit (PEF) and employee turnover intention (TI). The data obtained from 400 SME employees in Thailand were analyzed using SEM and MGA techniques. 

This research reveals many novel results. First, when analyzing the whole respondents using SEM, the findings indicate that person–supervisor fit (SPF), person–organization fit (POF), and person–job fit (PJF) positively influence PEF, thus discouraging TI. However, person–group fit (PGF) negatively affects PEF, which opposes the research framework and raises an issue for further analysis with the MGA approach. After analyzing the data of Generations X and Y separately using MGA, we found many interesting results. As for Generation X employees, PSF and PGF are not significantly related to PEF. However, POF and PJF remain the only two significant determinants improving Generation X’s PEF, which helps reduce TI. However, POF is more potent in influencing PEF than PJF. As for Generation Y employees, three variables influence their PEF—PSF, PGF, and POF. The results demonstrate that PSF and POF positively affect PEF and reduce TI. Interestingly, PGF negatively influences PEF, which shows the particular behavior of Generation Y respondents. Among the three variables (PSF, PGF, and POF), POF plays the most significant role in determining PEF for Generation Y employees. 

In conclusion, this study’s results suggest that more job suitability but less group involvement may encourage Generation Y employees to keep working in the organization in the post-pandemic era. However, job suitability may convince Generation X employees to remain in their job position in the long term.

### 6.2. Implications

Based on what we derived from the research findings, the following recommendations were proposed for academicians, Generation X employees, Generation Y employees, and organizations in Thailand that employ Generation X and Generation Y employees. There are two parts to this section: theoretical and practical implications. 

#### 6.2.1. Theoretical Implications

There are several theoretical implications derived from this research. Based on the PEF theory developed by Kristof-Brown (1996) [27], this study fills the research gap conducted in the emerging economy by reexamining the research model proposed by Ketkaew et al. (2020) [16]. Additionally, with inconclusive results of the previous studies regarding generational diversity, supportive work environments, and turnover intentions when conducting research in different circumstances [31,35], this research also fills the research gap by adding the moderating effects of Generation X and Y to the research model and using empirical data from Thailand to analyze this relationship. In contrast to previous studies, the MGA approach from this research paper yields many noteworthy results.

Additionally, many novel findings can be highlighted. Compared between Generation X and Y, Generation X employees require less supervisor support than Generation Y. This is because, at the organizations, many Generation X employees are senior, whereas Generation Y employees are young and might not fully adapt themselves to their tasks and work environments. Interestingly, our novel finding demonstrates that Generation Y employees have less of a need to work in a group than Generation X. In other words, Generation Y employees prefer to work by themselves because they feel that problems may arise when working in groups and that they cannot adapt themselves or make a final decision promptly. Lastly, the empirical results reveal that Generation X employees focus on the job’s suitability and require long-term jobs. In contrast, Generation Y employees may be inattentive about person–environment fit at workplaces and may leave that job regardless of the job suitability.

#### 6.2.2. Practical Implications

A customized approach is recommended for Generation X and Y employees and employers. Regarding person–supervisor fit, both Generation X and Generation Y employees can be dealt with in different ways. The first part is about the relationship between supervisors and employees; Generation X has no problem working without supervisors because they have already gained experience and know-how. Monitoring them closely with supervisors may negatively impact their work environment and may influence quitting intention. For Generation Y employees, due to a lack of understanding of and experiences in the organization, supervisor support can be essential. Supervisors can make sure that they know and understand what they need to do at the workplace. This approach may help to reduce mistakes and anxiety, which negatively affect their job performance.

As for Generation Y, the results reveal that person–group fit negatively influences the work environment and increases an employee’s intention to leave his/her job. People in Generation Y are relatively more individualistic than Generation X, especially those living in urban areas. To deal with Generation Y employees, although supervisory support is required, it is suggested the supervisor should monitor them at a distance and allow them to handle the task creatively. Utilizing technology in communication, training, and performance management may enable the supervisor to keep track of the Generation Y employee while giving him/her the liberty to work individually. However, clear job instructions and a thorough job orientation are required for Generation Y to effectively achieve organizational goals, in turn enhancing person–environment fit and reducing job turnover in the organization. 

The last part is about the organization, job, and environment. We can see that these three factors can help an organization retain Generation X and Generation Y employees. Management needs to increase their relationships by explaining their goals and values and clearly stating the required skills, knowledge, and ability. It is also suggested that management enhance the organization’s positive work environment to make sure that employees know and understand the purpose of the organization, thus making them more likely to remain in their jobs.

Finally, it is suggested that businesses try not to let age be a barrier in managing the company. In other words, they should avoid making age-based assumptions and stereotyping. Management should examine employees’ abilities and experience in order to determine what they can contribute to the role. They are old (or young) enough to handle the duty if they are good enough. Improving communication and establishing a mentorship program were effectively proven to help retain talented employees.

### 6.3. Limitations and Future Research

Several limitations exist in this study. First, our findings might not apply to all employees, as this research collected data from Generation X and Y, while there are some organizations in Thailand that still employ a proportion of the Baby Boomer generation. Second, sampling and data collection were restricted to Thailand. Therefore, caution is needed when generalizing the results. Moreover, this research employs data that cannot be used to analyze the employees’ behavior over a period of time. This limits the researchers’ ability to collect the actual employee turnover rate, leaving only the turnover intention variable to be considered. Hence, readers should be cautious when interpreting the research results. 

Nevertheless, it is worth noting that the proposed employment turnover model may be tested on a sample of people from various parts of the country and the world. This model could also be investigated in more depth, for example, by using a moderating effect to evaluate behavioral variations between generations by adding Generation Z to the analysis. Moreover, it is interesting to investigate the moderating roles of job positions and occupations.

## Figures and Tables

**Figure 1 behavsci-13-00489-f001:**
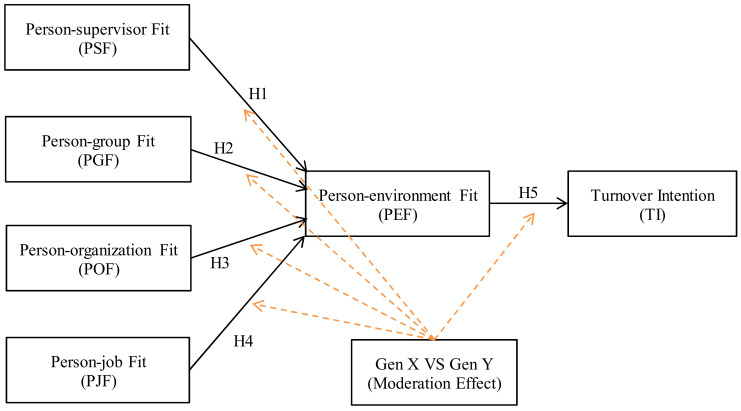
Proposed Model. Source: Figure created by authors, 2022.

**Figure 2 behavsci-13-00489-f002:**
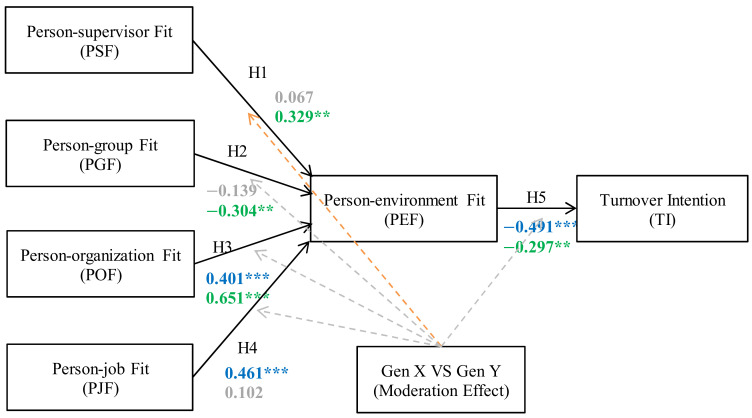
Multigroup Moderation Model. Source: Figure created by authors, 2023. *** denotes *p*-value < 0.001 significance level, ** denotes *p*-value < 0.05 significance level, blue texts denote Gen X, green texts denote Gen Y, and grey texts and arrows denote insignificance.

**Table 1 behavsci-13-00489-t001:** Demographic profile of respondents (n = 400).

Demographic Variables	Categories	Amount	Percentage
Gender	Male	166	41.50
	Female	234	58.50
	Total	400	100
Age	Generation X (43 to 57 years old)	294	73.50
	Generation Y (27 to 42 years old)	106	26.50
	Total	400	100
Education	High school	120	30
	Bachelor’s degree	235	58.75
	Graduate degree	35	8.75
	Diploma	10	2.50
	Total	400	100

Source: Data adapted from authors, 2023.

**Table 2 behavsci-13-00489-t002:** Goodness of fit of the measurement model.

Fit Indices	Value	Threshold	Assessment
Chi-Square	512.633		
df	237		
*p*-value	0.000		Acceptable for complex model
Chi-Square/df	2.163	≤5.00	Pass
CFI	0.945	≥0.90	Pass
IFI	0.945	≥0.90	Pass
TLI	0.936	≥0.90	Pass
RMSEA	0.055	≤0.08	Pass

Source: Values adapted from authors (2023). Thresholds gathered by [96], including chi-square/df threshold adapted from [77,90]; CFI from [77,90]; IFI from [77]; TLI from [77]; RMSEA from [77,97].

**Table 3 behavsci-13-00489-t003:** Convergent validity.

Indicator	Construct	Estimate	*p*-Value	Cronbach Alpha	AVE	CR
PSF1	PSF	0.84	***			
PSF2	PSF	0.787	***			
PSF3	PSF	0.745	***			
PSF4	PSF	0.818	***	0.875	0.637	0.875
PGF1	PGF	0.749	***			
PGF2	PGF	0.723	***			
PGF3	PGF	0.652	***			
PGF4	PGF	0.72	***	0.804	0.507	0.804
POF1	POF	0.652	***			
POF2	POF	0.831	***			
POF3	POF	0.836	***			
POF4	POF	0.818	***	0.868	0.621	0.866
PJF1	PJF	0.826	***			
PJF2	PJF	0.826	***			
PJF3	PJF	0.776	***			
PJF4	PJF	0.681	***	0.892	0.673	0.892
PEF1	PEF	0.556	***			
PEF2	PEF	0.689	***			
PEF3	PEF	0.748	***			
PEF4	PEF	0.797	***	0.798	0.497	0.796
TI1	TI	0.87	***			
TI2	TI	0.712	***			
TI3	TI	0.919	***			
TI4	TI	0.88	***	0.912	0.721	0.911

Note: *** denotes significant at <0.001. Source: Data adapted from authors, 2023.

**Table 4 behavsci-13-00489-t004:** Discriminant validity.

	TI	PEF	PJF	POF	PGF	PSF
TI	**0.849**					
PEF	−0.338	**0.705**				
PJF	−0.429	0.657	**0.82**			
POF	−0.391	0.672	0.677	**0.788**		
PGF	−0.12	0.304	0.48	0.517	**0.712**	
PSF	−0.223	0.491	0.521	0.513	0.354	**0.798**

Source: Data adapted from authors, 2023.

**Table 5 behavsci-13-00489-t005:** Goodness of fit of the structural model.

Fit Indices	Value	Threshold	Assessment
Chi-Square	541.185		
df	241		
*p*-value	0.000		Acceptable for complex model
Chi-Square/df	2.246	≤5.00	Pass
CFI	0.940	≥0.90	Pass
IFI	0.941	≥0.90	Pass
TLI	0.931	≥0.90	Pass
RMSEA	0.057	≤0.08	Pass

Source: Values adapted from authors (2023). Thresholds gathered by [96], including chi-square/df threshold adapted from [77,90]; CFI from [77,90]; IFI from [77]; TLI from [77]; RMSEA from [77,97].

**Table 6 behavsci-13-00489-t006:** Hypothesis test results from the structural model.

Hypothesis	Endovenous Variable	Exogenous Variable	Standardized Estimate	*p*-Value	Result
H1	PSF	PEF	0.122	0.039 **	Supported
H2	PGF	PEF	−0.153	0.015 **	Contradicted
H3	POF	PEF	0.443	>0.001 ***	Supported
H4	PJF	PEF	0.39	>0.001 ***	Supported
H5	PEF	TI	−0.394	>0.001 ***	Supported

Note: *** denotes significant at <0.001 and ** denotes significant at <0.05. Source: Data adapted from authors, 2023.

**Table 7 behavsci-13-00489-t007:** Measurement invariance (Generation X vs. Y).

Fit Indices	Configural Invariance	Metric Invariance	Scalar Invariance	Threshold
*p*-value	0	0	0	
Chi-Square/df	1.691	1.679	1.639	≤5.00
CFI	0.935	0.934	0.935	≥0.90
IFI	0.936	0.935	0.936	≥0.90
TLI	0.925	0.926	0.930	≥0.90
RMSEA	0.043	0.042	0.041	≤0.08

Source: Values adapted from authors (2023). Thresholds gathered by [96], including chi-square/df threshold adapted from [77,90]; CFI from [77,90]; IFI from [77]; TLI from [77]; RMSEA from [77,97].

**Table 8 behavsci-13-00489-t008:** Test results for loading differences.

Path	Relationship	Standardized Loading		Critical Ratio Difference	Threshold
		Gen X	Gen Y		
H1	PSF --> PEF	0.067	0.329 **	2.052 **	1.96
H2	PGF --> PEF	−0.139	−0.304 **	−1.518	−1.96
H3	POF --> PEF	0.401 ***	0.651 ***	1.156	1.96
H4	PJF --> PEF	0.461 ***	0.102	−1.474	−1.96
H5	PEF --> TI	−0.419 ***	−0.297 **	1.38	1.96

Note: *** denotes *p*-value < 0.001 significance level, ** denotes *p*-value < 0.05 significance level. Source: Data adapted from authors, 2023.

## Data Availability

Data is contained within the article.

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
