# Peer review of "An Analysis of Multigenerational Issues of Generation X and Y Employees in Small- and Medium-Sized Enterprises in Thailand: The Moderation Effect of Age Groups on Person–Environment Fit and Turnover Intention"

_behavsci, 2023, doi:10.3390/bs13060489_

Round 1

Reviewer 1 Report

The presented paper is interesting in its design and practical use and is a contribution to science. Despite its indisputable qualities, I have a few comments:

• Despite the generally known terms, do not use abbreviations in the title and abstract (SME, SEM, MGA), but only use them in the introduction and with a conceptual explanation.

• Is your statistical sample relevant? What is the minimum size of the ensemble given the size of the finite population and the chosen error of estimate?

• Retention of a person in the workplace is also supported by the right motivation (mainly for employees of generation X. This makes it possible to reduce the fluctuation coefficient. I recommend adding a short section dedicated to this issue (see e.g. DOI 10.21676/23897848.4907, DOI 10.3846/jbem.2020.13702, DOI 10.17512/pjms.2021.23.1.14)

• Despite the high number of used literature, it is mostly outdated (up to 2/3 of the literature is older than 5 years). I recommend adding newer literature from the last 5 years.

Take my comments as a recommendation to improve the level of your work. I wish the authors much success in their further scientific work.

Minor editing of English language required

Author Response

Dear Reviewer,

Thank you very much for your valuable comments.

Please see the responses in the attached file.

Best,

Author

Reviewer 2 Report

Dear Authors,

 I am sending you my review as an attachment.

Best regards

Author Response

Dear Reviewer,

Thank you for your valuable comments.

Please see the responses as attached.

Best,

Author

Round 2

Reviewer 1 Report

I thank the authors for accepting comments and wish them success in their further scientific work.